# Life Cycle Environmental Impact Assessment and Applicability of Synthetic Resin Formwork

**DOI:** 10.3390/ma16020696

**Published:** 2023-01-10

**Authors:** Kyung-Yong Nam, Myung-Kwan Lim

**Affiliations:** 1Architectural Technology Team, Taeyoung E&C Co., Ltd., Engineering & Construction Group, Seoul 07241, Republic of Korea; 2Department of Architecture Engineering, University of Ulsan, Ulsan 44610, Republic of Korea

**Keywords:** synthetic resin formwork, Euro form, environmental impact, life cycle assessment, CO_2_ emissions

## Abstract

In this study, the environmental impacts of the production and field application of synthetic resin formwork were quantitatively compared to the Euro form. The noise test results showed an average of 107.3 dB (A) for the Euro form and 99.7 dB (A) for the synthetic resin formwork. Additionally, when the number of uses was considered, the CO_2_ emissions from the synthetic resin formwork were approximately 32% lower than the Euro form. Based on these results, it is expected that the use of synthetic resin formwork will reduce material production by half and reduce CO_2_ emissions compared to channel formwork.

## 1. Introduction

### 1.1. Purpose of the Study

For high-rise buildings with reinforced concrete (RC) structures, the selection of an appropriate formwork represents a considerable proportion of the total cost of RC buildings [1,2], as it is an important factor in the structure’s cycle time per floor [3].

This is because the selection of an appropriate formwork affects the amount of time required for construction as well as the processes required for electricity, machinery, equipment, and finish work [4].

In addition, the selection of an appropriate formwork can increase work efficiency for concrete finishes and subsequent processes.

In the South Korea, some residential buildings have lightweight steel frame structures, wooden structures, or masonry structures, but most buildings are constructed with an RC structure. This has resulted in a high demand for products that are involved in the formwork process [1].

The development of conventional (small) formwork, which is required in small and medium construction sites, has been delayed or halted entirely because of market conditions. Metal formwork is expensive, heavy, and causes a large amount of noise during construction. The Euro form, which is made of metal and wood, is also heavy and difficult to recycle, but it is inexpensive compared to entirely metal products. Wooden formwork also has problems; it must be constructed at the site, and not only does its construction generate a large amount of wood debris, but also, the recycling rate of the remaining waste material is low. This problem, as well as the forest damage caused by the demand for wood, must be addressed worldwide [5].

On one hand, wooden formwork has been commonly used in the construction of concrete structures around the world because of its versatility, ease of handling, and low cost. Due to these factors, many construction companies prefer it. However, the use of wooden formwork must be gradually decreased due to the environmental destruction caused by the increased demand for wood [5].

On the other hand, the Euro form generates a large amount of noise during the installation and dismantling processes. This is because its frame is made of metal, and the impacts of tools (e.g., hammers) produce noises. Because repeated impact noises from construction sites cause adverse physical and mental effects and increased stress in nearby residents, measures to reduce the amount of such noise are urgently required [6,7].

To address these problems, plastic formwork based on synthetic resins has been developed. Synthetic resin formwork is made using a single raw material, and no further production is required at the site because the panel and frame are produced in one piece. In contrast, the plywood used for wooden formwork is mostly decayed or discarded after use. Thus, synthetic resin formwork is more economically efficient because it has higher overall usability than plywood formwork or the Euro form.

Formwork is mostly rented for a certain period of time to perform specific concrete work. Therefore, it is possible to reduce formwork rental costs, and thereby save construction costs, by establishing thorough construction plans [8].

After synthetic resin formwork has been used, 100% of it can be recycled through pulverization. This is highly beneficial, as the total recovery of waste and recycled materials at a low cost is an essential requirement for modern society.

Preventing global warming is also important, and global efforts to improve energy efficiency and reduce carbon emissions require an understanding of how different industries contribute to the total amount of emissions [9]. For example, in Brazil, carbon dioxide emissions from the construction industry represent 23% of the country’s total carbon dioxide emissions [10].

Additional pollutants are also generated by the construction industry, including carbon dioxide (CO_2_), carbon monoxide (CO), methane (CH_4_), nitrogen oxides (NOX), sulfur dioxide (SO_2_), and nitrous oxide (N_2_O). These contribute to global warming, acidification, eutrophication, and ozone depletion [11].

South Korea has the ninth-highest rate of carbon dioxide emission in the world [12]. In response to the increasing number of environmental problems caused by the construction industry, various studies have been conducted by the Korean government. In particular, the environmental impacts of the construction, operation, and demolition life cycles have been investigated [13,14]. Even though there have been many studies conducted on building life cycle assessment (LCA) [15,16,17,18,19,20,21], the assessment of the environmental impacts of building materials remains insufficient.

To reduce the environmental impacts from the construction industry, the environmental impacts of each factor in the construction of buildings must first be quantified [13].

The energy consumption and carbon emissions associated with the construction of buildings are separated into five stages: material production, construction, operation, demolition, and material disposal and recycling. Environmental impact assessments have been based mostly on the building life cycle, and environmental impact assessments of building material production and construction have rarely been conducted [12].

Therefore, this study focuses on examining the latter. Specifically, the purpose of this study is to examine the applicability of recyclable synthetic resin formwork and to quantitatively compare the environmental impacts of the production and field application of synthetic resin formwork to those of the Euro form. The formwork type dealt with in this study is not a large system formwork, such as the auto climbing system (ACS), climbing form, and gang form; it is small unit formwork, which is similar to Euro forms or wooden formworks that are commonly used at small and medium sites in residential areas rather than high-rise construction sites in urban areas.

### 1.2. Cautions for Selecting Formwork Materials

The following criteria must be satisfied to ensure the quality and workability required for formwork [22,23]:
(1)The material must be easy to work with using manpower and tools, and it must not be destroyed or cracked during nailing.(2)It must be durable enough to withstand the impact of the contact surface in normal conditions, such as fixing steel reinforcement, fixing the formwork itself, and concrete pouring.(3)It must be light enough for formwork workers to install and transport. It also must be strong enough to withstand the considerable loads and impacts that may occur during concrete pouring.(4)It must be stable when exposed to direct sunlight, snow, or rain, and it must be able to withstand excessive twisting and to resist swelling.(5)It must not excessively absorb moisture from the concrete after it has been poured.

### 1.3. Research Method and Scope

The scope of this study was limited to the Euro form and synthetic resin formwork types. Experiments were conducted to compare the noise that occurred during the installation and dismantling of both types of formwork and to perform a visual inspection of the formwork deformation that may occur during concrete pouring. The “Korean environmental impact assessment index methodology” produced by the Korean Ministry of Environment (ME) [24] was used. In addition, for the LCA of the synthetic resin formwork, a process flow diagram was utilized to satisfy the system boundary required by ISO FDIS 1335-2 [25]. The following describes the research method used in this study:(1)Synthetic resin formwork was mainly applied to small and medium sites in residential areas, not in urban areas. Tests on the evaluation of the physical properties and material performance of synthetic resin formwork in the existing literature were verified [26,27,28].(2)A noise test was conducted to compare the intensity of the noise generated during the construction of synthetic resin formwork to that of the Euro form.(3)The constructability of the synthetic resin formwork constructed on-site was investigated, and its benefits were compared to those of the Euro form.(4)The environmental impact of synthetic resin formwork during its production life cycle was evaluated.

## 2. Materials and Methods

### 2.1. Synthetic Resin Formwork

Figure 1 shows a sample of high-density polyethylene (HDPE), a material that can produce synthetic resin formwork through injection into a form. The synthetic resin formwork used in this study was produced by injecting HDPE into a dedicated form, as shown in Figure 2. Color can be added to the formwork by combining pigment grains with the HDPE. HDPE was discovered in the 1950s after low-density polyethylene (LDPE) was first discovered by researchers in ICI in England and after the development of linear low-density polyethylene (LLPE) in the 1930s [29]. HDPE is also referred to as low-pressure polyethylene because the polymerization of HDPE is performed under a lower pressure than that of LDPE using a catalyst. It is also referred to as hard polyethylene because its stiffness is higher than that of LDPE. Table 1 lists the physical and mechanical properties of the HDPE used in this study.

### 2.2. Synthetic Resin Formwork Production Process

In the case of the Euro form, which was used for comparison in this study, the final product is completed when the cut and coated plywood is combined with the steel frame after it is processed, welded, and painted. Unlike the Euro form, synthetic resin formwork is produced using only a single material (i.e., HDPE). There are large differences between the Euro form and synthetic resin formwork in terms of the amount of work, input manpower, and time required for the production processes.

Figure 3 depicts the production process for synthetic resin formwork. First, HDPE grains with diameters between 3 and 5 mm are placed into an injection molding machine, and the settings for the injection process are set. Based on the settings, a formwork injection is performed to produce the finished product. It takes approximately two minutes for the raw material to be converted into the finished product. This production process has many benefits compared to the Euro form in terms of the process type, production time, and stability [26,27,28]. However, the postinjection cooling process must not be altered because attempting to reduce cooling time to shorten production time may cause deformation of the formwork.

## 3. Applicability Evaluation

### 3.1. Noise Test

#### 3.1.1. Experimental Plan and Method

Table 2 shows the experimental plan for the noise test, and Figure 4 shows the Euro form and synthetic resin formwork used in the test. The test was conducted outdoors in conditions similar to typical field conditions. The noise generated when an impact is applied to the edge of the formwork during assembly and demolding was measured for both the Euro form and synthetic resin formwork. To generate the noise, an impact was directly applied by the person in charge of an authorized testing agency in Korea (Korea Conformity Laboratories), as shown in Figure 5. To measure the noise, the Solo 01dB-Metravib (Acoem, Limonest, France) was used as a sound level meter, and the CAL02 01dB-Stell (Japan) was used as a sound level calibrator. When an impact was applied to the center of the metal formwork frame with a hammer, the maximum noise level was measured five times at a location 1.5 m in front of the formwork. During the measurement, the A-weighting network was applied as a fast dynamic characteristic mode.

#### 3.1.2. Noise Measurement Results

The results of the noise test on the Euro form and synthetic resin formwork are presented in Table 3. When an impact was applied five times to the specimens, the Euro form exhibited an average of 107.3 dB, and the synthetic resin formwork produced an average of 99.7 dB. The difference between the two specimens was 7.6 dB, which is significant considering the measurement distance of 1.5 m and the fact that a metal hammer was used for the impacts. This difference was even larger when the results of the noise characteristics were analyzed in terms of frequency. Thus, the synthetic resin formwork generated less noise than the Euro form. Because the experimental results may vary depending on the measurement conditions, the final results for any given application must be derived through multiple tests rather than only one.

#### 3.1.3. Noise Characteristics as a Function of Frequency

Figure 6 shows the noise characteristics of the Euro form and the synthetic resin formwork as a function of frequency. The maximum sound pressure level was 86.8 dB (A) for the Euro form and 77.4 dB (A) for the synthetic resin formwork. When the sound pressure level was measured according to the frequency, irregular patterns were observed in the low-frequency band between 31.5 and 200 Hz. The peak values of 79 (A) and 86.8 dB (A) were reached at 800 Hz and 3.15 kHz, respectively. After the peak values were reached, the noise level gradually decreased. In the 31.5–200 Hz range, which exhibited irregular patterns, the two formwork types showed similar dB values, but they exhibited significantly different patterns in the high-frequency region beyond 800 Hz. In the region above 1 kHz, the difference in the sound pressure level between the synthetic resin formwork and the Euro form became much larger, indicating that the noise fatigue of the Euro form was higher than that of the synthetic resin formwork at high frequencies. The literature indicates that the sound pressure level tends to have irregular patterns in the low-frequency range and tends to attenuate in the high-frequency band [29,30,31]. This appears to be because lower frequencies with higher energy experience less acoustic energy attenuation compared to higher frequencies [29]. The noise characteristics of the Euro form, as measured in this study, are expected to have adverse physical and mental effects on nearby residents if its high sound pressure level is continuously transmitted to residential areas near construction sites [29,30]. Complaints about construction site noise are one of the most serious social problems in the South Korea.

### 3.2. Application of Synthetic Resin Formwork

The synthetic resin formwork was constructed at a construction site to examine its field applicability. It had a size of 300 mm × 1200 mm and a weight of 4.06 kg, which was one-third the weight of the Euro form (which was also 300 mm × 1200 mm). A typical concrete mixture that was produced and readily available in the local area was used. Because the physical properties and fluidity of the concrete had insignificant impacts on the experimental results of this study, their detailed descriptions have been omitted. Figure 7 shows the connection between the Euro form and the synthetic resin formwork. The latter had the same connection configuration as the former, including flat ties and wedge pins. Their construction methods were also the same.

In addition, when nails or other hardware were applied to the synthetic resin formwork, the formwork surface was not broken or cracked. Therefore, it exhibited higher workability during installation than the Euro form. The noise test results showed that the synthetic resin formwork caused less worker fatigue by generating less noise while it was being processed.

Figure 8 shows the concrete poured into the synthetic resin formwork. For slab formworks, concrete pouring can be visually examined through the construction of synthetic resin products dedicated to slabs. However, in the case of vertical walls, it is difficult to visually examine the status of concrete pouring. Because the compatibility of concrete is structurally more important for vertical walls than slabs, the status of concrete pouring for vertical structures can be examined in real time through the application of synthetic resin formwork. Synthetic resin formwork is also favorable for efficiently capitalizing on the remaining concrete because the status of the concrete pouring can be immediately determined.

Figure 9 shows the concrete surface after removing the Euro form and the synthetic resin formwork. The Euro form left a number of stains and pores even though form oil was applied to the formwork surface. For the synthetic resin formwork, to which oil was not applied, defects that typically occur on the concrete surface (e.g., stains, pores, dusting, and rock pockets) were not found. The concrete surface is significantly affected by the degree of compaction in the concrete pouring process, but the finish of the concrete surface after formwork removal may vary according to the formwork surface material and storage conditions [5]. In addition, unlike the conventional formwork removal method in which impacts with field tools (e.g., hammers and metal materials) are used to dismantle the formwork, simple bending without tools can just as easily facilitate dismantling. This can minimize damage to structures caused by impacts and vibrations [23]. In general, the concrete surface must be smooth and have a uniform color. The most important factors in achieving this are formwork joint construction and quality management [32,33]. Therefore, whether or not the formwork materials can be recycled depends significantly on the quality of the components used for formwork production [34]. Finally, vague lattice marks appeared on the surface of the synthetic resin formwork concrete, which occurred because the formwork panel was thin. This was rectified by adjusting the panel thickness.

## 4. Procedure for Life Cycle Environmental Impact Assessment

### 4.1. Assessment Purpose and Scope Setting

The life cycle impacts of the synthetic resin formwork materials were assessed in accordance with the assessment methodology of ISO 14040 [35], an international standard for LCA. Product category rules (PCR) for building materials were prepared separately for each country.

The purpose of the LCA performed in this study was to construct a life cycle inventory (LCI) database (DB) for synthetic resin formwork and to quantitatively compare the environmental impact assessments of synthetic resin formwork with those of the conventional Euro form. The main categories of life cycle environmental impacts include resource depletion, global warming, ozone layer destruction, acidification, eutrophication, and photochemical oxide production [12]. The functional unit of the synthetic resin was expressed in terms of one piece (EA), and the amount of material used for production was based on one product with the dimensions of 300 mm × 1200 mm. In addition, the weight of one (EA) synthetic resin formwork was 4.3 kg, which corresponded to the amount of the HDPE used. One (EA) Euro form of the same size was used as a comparison. The weight of the Euro form was 12.8 kg (the plywood contributed 2.8 kg, and the steel frame contributed 10 kg).

### 4.2. Boundary Conditions

Figure 10 shows the process flow diagram of the synthetic resin formwork, and Figure 11 shows its system boundary. The categories of air emissions, water emissions, and waste from raw materials, energy, and auxiliary materials were considered from the raw material production stage to the stage just before product shipment. In terms of the data quality of the production of synthetic resin formwork, the regional boundary was the synthetic resin formwork production factory located in Gwangju, Korea, the temporal boundary was the data being limited to the products produced in 2020, and the technical boundaries were indoor production and the process technology standards of the synthetic resin formwork production company and their injection factory.

### 4.3. LCI Data Analysis

LCI data analysis is the process of calculating environmental loads by recording the types and amounts of input elements (including HDPE, plywood, steel, and epoxy paint) and output elements (i.e., synthetic resin formwork and the Euro form) for the system boundaries set in the research scope. Therefore, LCI data on the materials required for the production of synthetic resin formwork and the Euro form in terms of the LCA, mass and areas of the materials as well as transport of the produced materials to the factory are required. Table 4 shows the LCI DB used in the process shown in Figure 10. For LCI data, it is desirable to use the basic units provided by each country because temperature, energy, and natural resources differ according to the country. In the absence of process-based LCI data from previous studies, “surrogate LCAs” were used to approximate the options closest to the components [36]. For example, steel produced in the South Korea can be used as an approximation of steel produced in Japan. The data provided by ME [24,37] and the Ministry of Land, Infrastructure, and Transport in Korea were used for the LCI data, and the Total software program was used for LCI data analysis. HDPE was set as the coupling material for the synthetic resin formwork, whereas plywood, a steel frame, and epoxy paint were set as the coupling materials for the Euro form. For transport information conditions, the two conditions of trucks weighing over 12 tons and traveling a distance of 50 km were applied in the same manner.

### 4.4. CO_2_ Evaluation Procedure

The amount of CO_2_ (*C_d_*) emitted from the system during the production of each raw material for the formwork functional unit can be calculated using
(1)$$C_{d}=CO_{2\mathrm{-M}}+CO_{2\mathrm{-T}}+CO_{2\mathrm{-p}}$$
where *CO*_2-*M*_ is the CO_2_ emissions produced in the material stage (which includes high-density synthetic resin and plywood), *CO*_2-*T*_ is the CO_2_ emissions produced in the transport stage from the outlet of each material to the formwork production factory, and *CO*_2-*P*_ is the CO_2_ emissions produced in the factory production stage. The quantity *CO*_2-*M*_ can be calculated using
(2)$$CO_{2\mathrm{-M}}=\overset{n}{\underset{i=1}{\sum }}(W_{i}\times CO_{2\left(i\right)\mathrm{-L}CI})\;$$
where *i* is the material used for formwork production, n is the number of such materials, and Wi and *CO*_2*(i)-LCI*_ are the usage (kg) and (*CO*_2-*kg/kg*_) of material *i*, respectively.

### 4.5. Environmental Impact Assessment

The potential environmental impacts based on the LCI DB results were assessed through classification (shown in Table 5) and weighting (shown in Table 6) processes to exclude subjective aspects and quantify the magnitudes of environmental impacts [11]. The magnitude of the environmental impact of the synthetic resin formwork (WIi) was quantified using
(3)$$WI_{i}=\sum_{i}\frac{C}{N_{i}}\frac{N_{i}}{T_{i}}f_{i}=\sum_{i}\frac{CI_{i}}{N_{i}}W_{i}=\frac{\sum_{j}(Load_{i}*eqv_{i,j})}{N_{i}}W_{i}$$
where *CI_i_* is the magnitude of the impact of all inventory items (*j*) included in impact category *i* on the impact category they belong to, *N_i_* is the normalization reference of impact category *i*, *W_i_* is the weight of impact category *i*, *Load_j_* is the environmental load of the *j*-th inventory item, and *eqv_i,j_* is the characterization coefficient value of the *j*-th inventory item that belongs to impact category *i*. Equation (3) is the environmental impact assessment index of this study, which was based on the equations suggested by ME [24,37] and Yang et al. [38].

Weighting methods may vary because they are a function of country, culture, and time as well as of social, political, and ethical standards for environmental impacts rather than of scientific facts. In this study, the values suggested by ME [24,37], which were calculated using the analytic hierarchy process based on sequence information, were used to model the reliability for the weights of each impact category.

Figure 12 shows the CO_2_ emissions generated during the production of the Euro form and synthetic resin formwork.

## 5. Life Cycle Environmental Impact Assessment of Synthetic Resin Formwork

### 5.1. CO_2_ Emissions of Synthetic Resin Formwork

The synthetic resin formwork generated 34% more CO_2_ emissions during production than the Euro form did. In general, the CO_2_ emissions generated by producing formwork were dominated by the materials used. The synthetic resin formwork emitted 8.1 kg/EA due to the HDPE (a highly durable material with an emission factor of 1.88 CO_2_-kg/kg), which was higher than the 6.9 kg/EA emitted by the Euro form. In terms of transport, the CO_2_ emissions produced were 1.5 × 10^−5^ kg/EA for the Euro form and 0.5 × 10^−6^ kg/EA for the synthetic resin formwork (these values are too small to be seen in Figure 12).

Figure 13 shows the CO_2_ emissions produced by the Euro form and the synthetic resin formwork if the number of uses is considered. Although the synthetic resin formwork produced more CO_2_ emissions during production than did the Euro form, the CO_2_ emissions from the synthetic resin formwork were 32% lower than those of the Euro form when the number of uses was considered. The number of uses for the Euro form is less than 15, which is half the number of uses for synthetic resin formwork. Because the use of raw materials and equipment for the Euro form is twice as high as that of synthetic resin formwork, the Euro form emits more CO_2_ than synthetic resin formwork in actual field applications. The transport stage exhibited the same CO_2_ emissions as the production stage.

### 5.2. Environmental Impact Assessment of Synthetic Resin Formwork

Table 7 and Figure 14 show the environmental impacts of the Euro form and synthetic resin formwork after the processes of classification, normalization, characterization, and weighting. The CO_2_ emissions were reevaluated to consider the results in Figure 12 and Figure 13. The number of uses for the synthetic resin formwork was assumed to be 30, and that of the Euro form was assumed to be 15. Because the number of uses may depend on the site, sufficient examination is required to confirm these values for any given application. The environmental load of the synthetic resin formwork was approximately 56% higher than the Euro form. However, when the number of uses was considered, the environmental load of the synthetic resin formwork was approximately 21% lower. The synthetic resin formwork produced a large amount of CO_2_ emissions in the production stage because of the characteristics of the material, but the CO_2_ emissions decreased as the number of uses increased. Based on 30 uses, the impacts of synthetic resin formwork on global warming, acidification, eutrophication, human toxicity, and resource depletion were estimated to be lower than the Euro form by 30, 38, 38, 38, and 25%, respectively. By reducing the use of the Euro form and increasing the use and recycling of synthetic resin formwork, the environmental destruction caused by logging can be mitigated [5].

## 6. Conclusions

In this study, the noise, construction impacts, and environmental impacts generated by the production and field application of the Euro form and synthetic resin formwork were quantitatively compared. The results are summarized as follows:(1)In the high-frequency range, there was a significant difference in sound pressure level between the synthetic resin formwork and the Euro form. This demonstrates that the noise fatigue of the Euro form was higher than that of the synthetic resin formwork at high frequencies. If the high sound pressure levels that come from the Euro form are continuously transmitted to residential areas, then they will have adverse physical and mental effects on nearby residents.(2)When the synthetic resin formwork was constructed in place of the Euro form, there were no compatibility problems. In addition, after the formwork was removed, the concrete surface finish of the synthetic resin formwork was superior to that of the Euro form even though no oil was applied to the synthetic resin formwork.(3)The synthetic resin formwork generated approximately 34% more CO_2_ emissions than the Euro form during production. This is because the CO_2_ emissions of formwork are generally dominated by the materials used, and the production of synthetic resin formwork generated higher CO_2_ emissions than the Euro form because of the HDPE, a corrosion-resistant material. However, when the number of uses for formwork was considered, the CO_2_ emissions produced by the synthetic resin formwork were approximately 32% lower than those of the Euro form.(4)The environmental impact load of the synthetic resin formwork, considering the number of uses, was 6.6 × 10^−7^ kg-eq/30 uses, which was approximately 21% lower than that of the Euro form (8.4 × 10^−7^ kg-eq/30 uses). In addition, based on 30 uses, the impacts of synthetic resin formwork on global warming, acidification, eutrophication, human toxicity, and resource depletion were found to be lower than the Euro form by 30, 38, 38, 38, and 25%, respectively.

Based on the results of this study, the applicability of synthetic resin formwork, as well as its on-site structural performance, will be further evaluated in a future study. Although structural safety was ensured through laboratory tests, various safety verification experiments are also required because there are many risk factors that may occur in the field.

## Figures and Tables

**Figure 1 materials-16-00696-f001:**
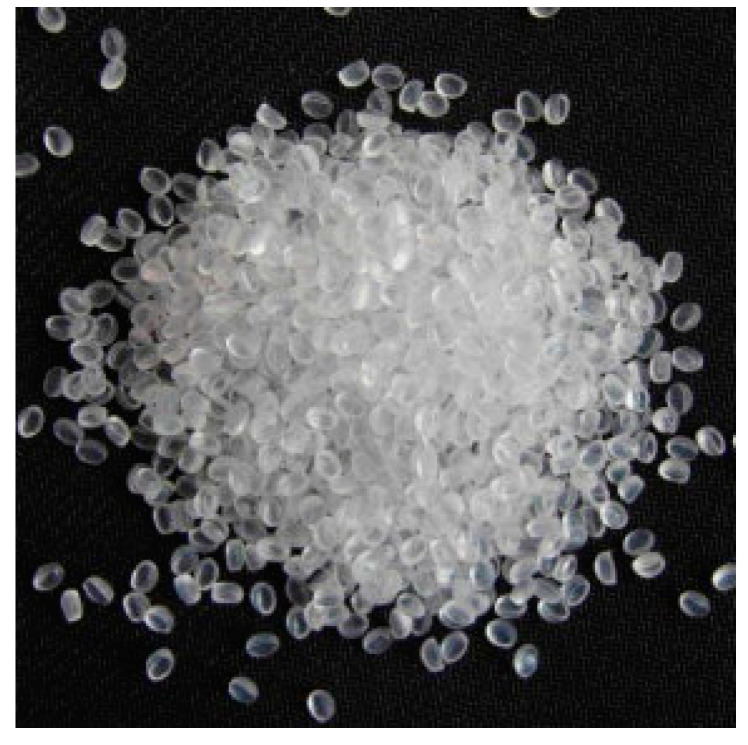
High-density polyethylene (HDPE).

**Figure 2 materials-16-00696-f002:**
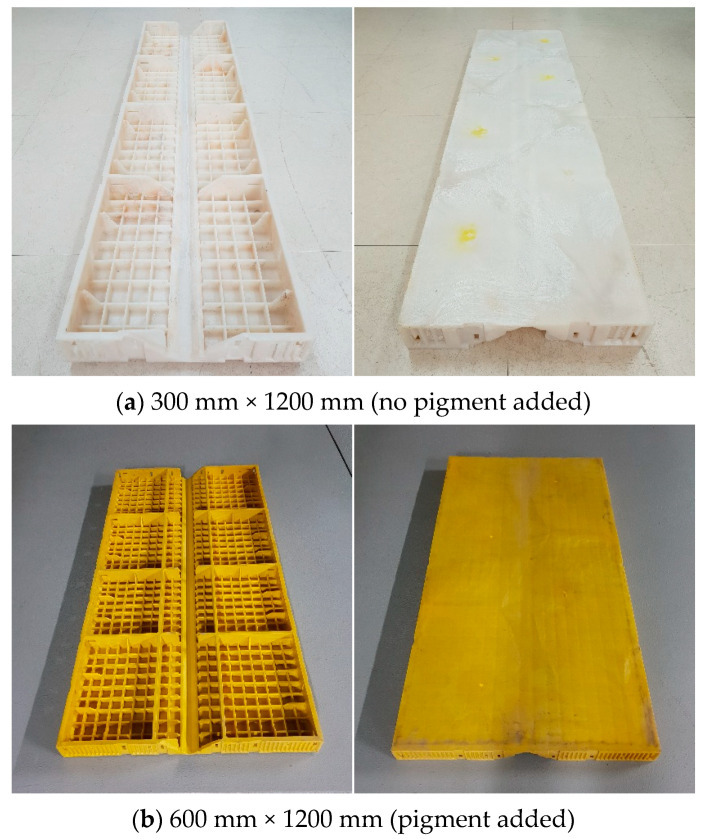
Synthetic resin form.

**Figure 3 materials-16-00696-f003:**
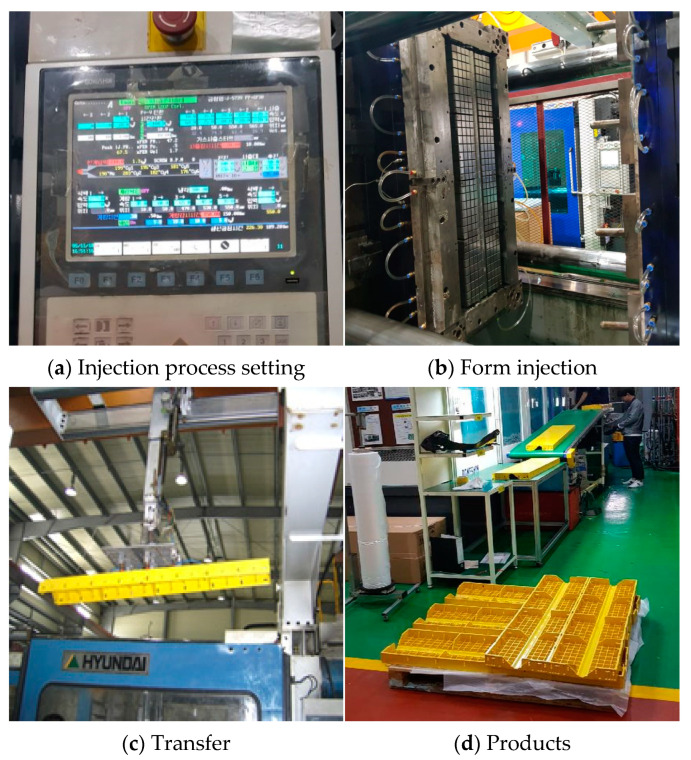
Synthetic resin forming process.

**Figure 4 materials-16-00696-f004:**
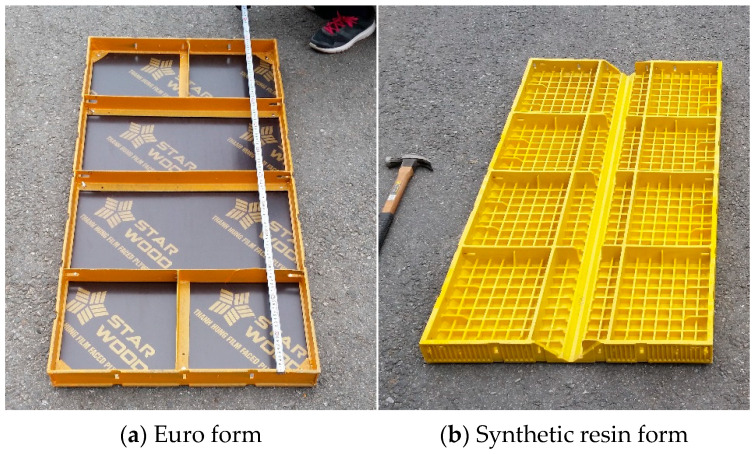
Noise test specimen.

**Figure 5 materials-16-00696-f005:**
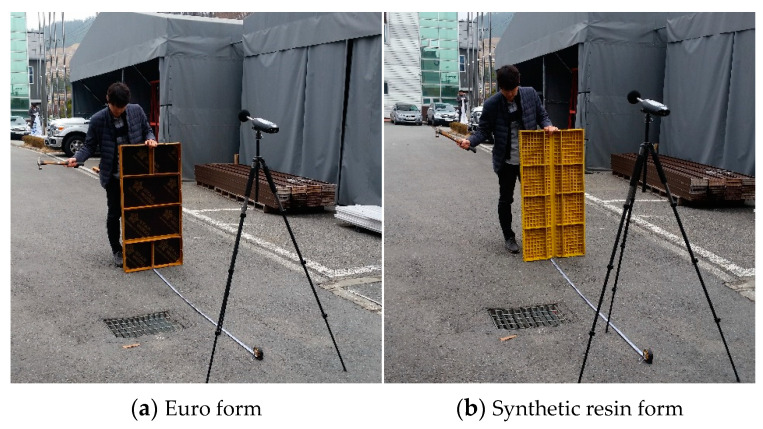
Noise test method.

**Figure 6 materials-16-00696-f006:**
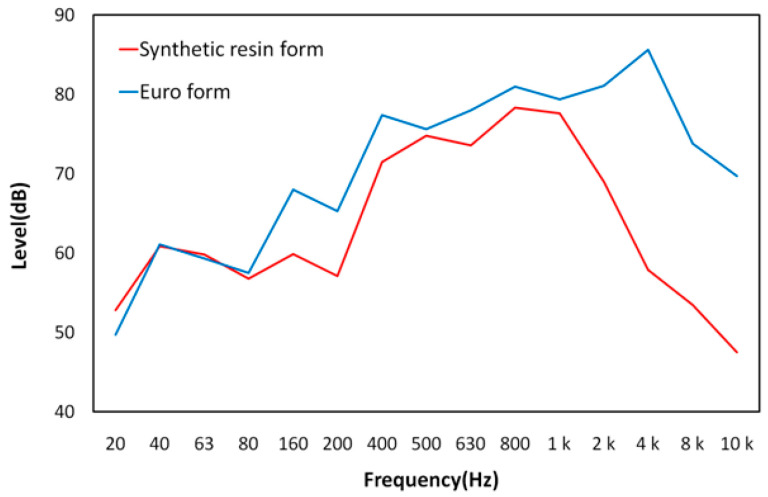
Frequency noise factor according to form type.

**Figure 7 materials-16-00696-f007:**
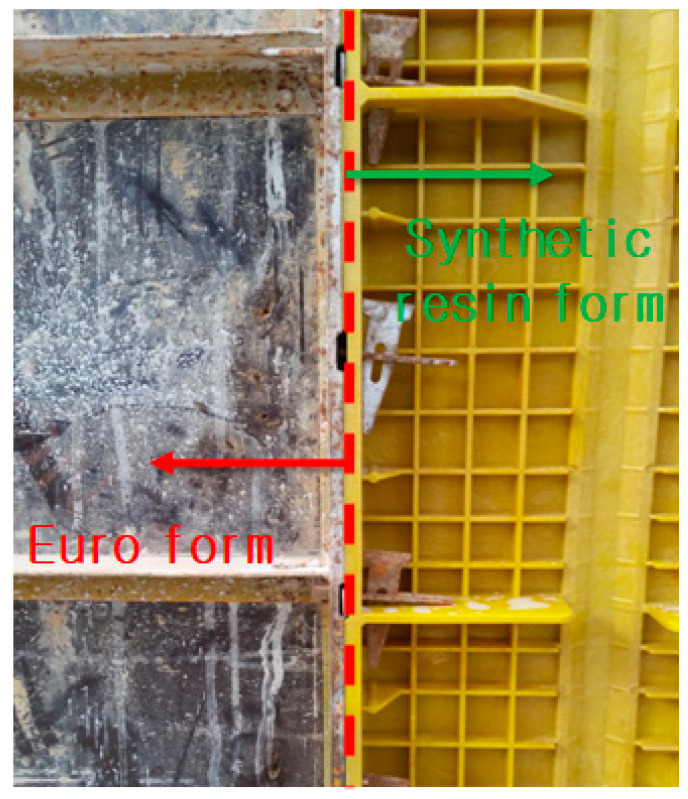
Connection between the Euro form and the synthetic resin formwork.

**Figure 8 materials-16-00696-f008:**
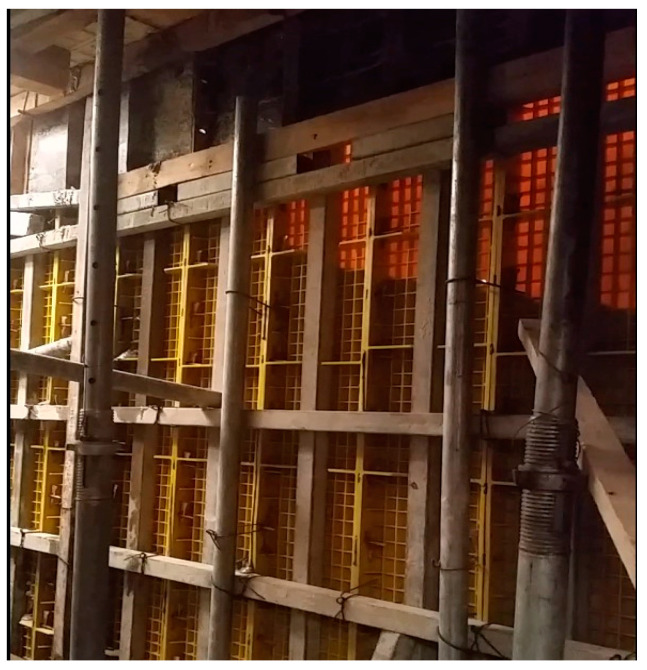
Concrete wall placing.

**Figure 9 materials-16-00696-f009:**
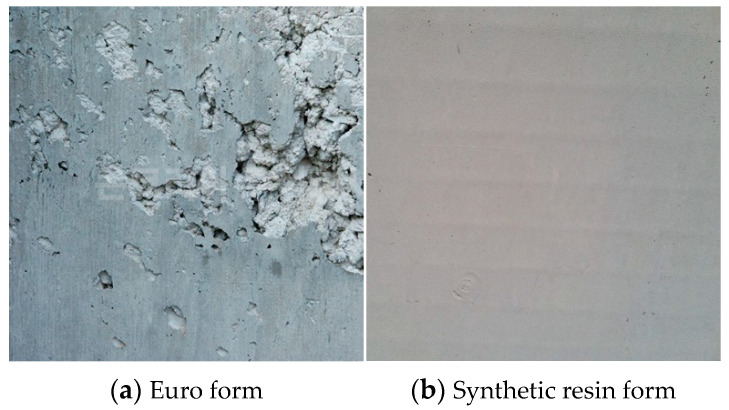
Concrete finish.

**Figure 10 materials-16-00696-f010:**
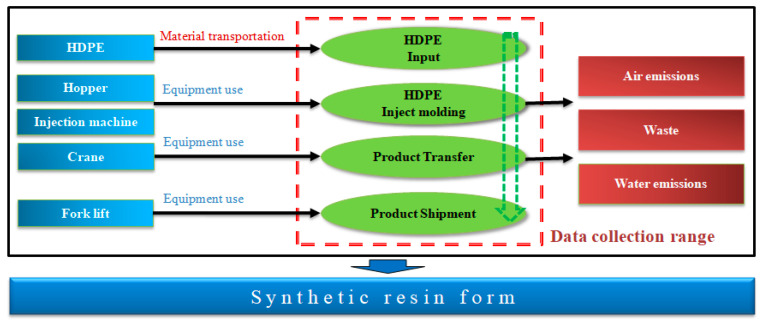
Process flow diagram of synthetic resin form production.

**Figure 11 materials-16-00696-f011:**
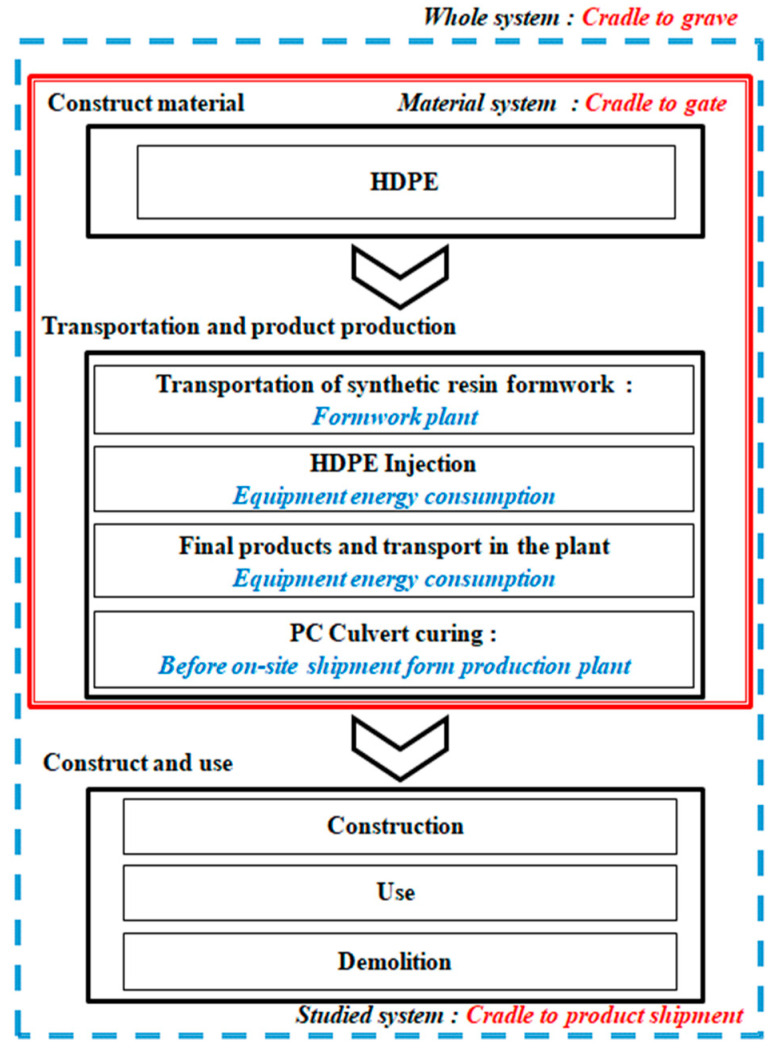
System boundary for synthetic resin form production.

**Figure 12 materials-16-00696-f012:**
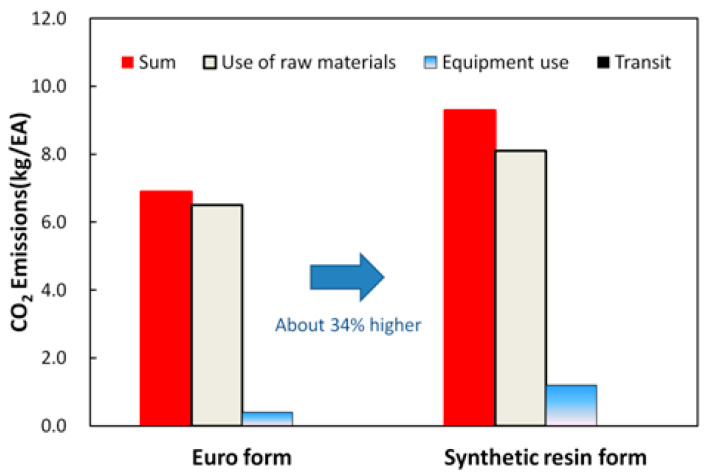
CO_2_ emissions generated during the production of the Euro form and synthetic resin formwork.

**Figure 13 materials-16-00696-f013:**
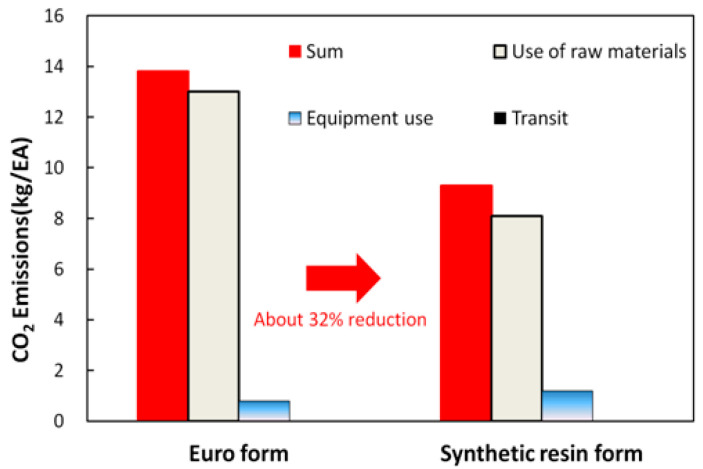
CO_2_ emissions generated during the production of the Euro form and synthetic resin formwork, accounting for the number of uses.

**Figure 14 materials-16-00696-f014:**
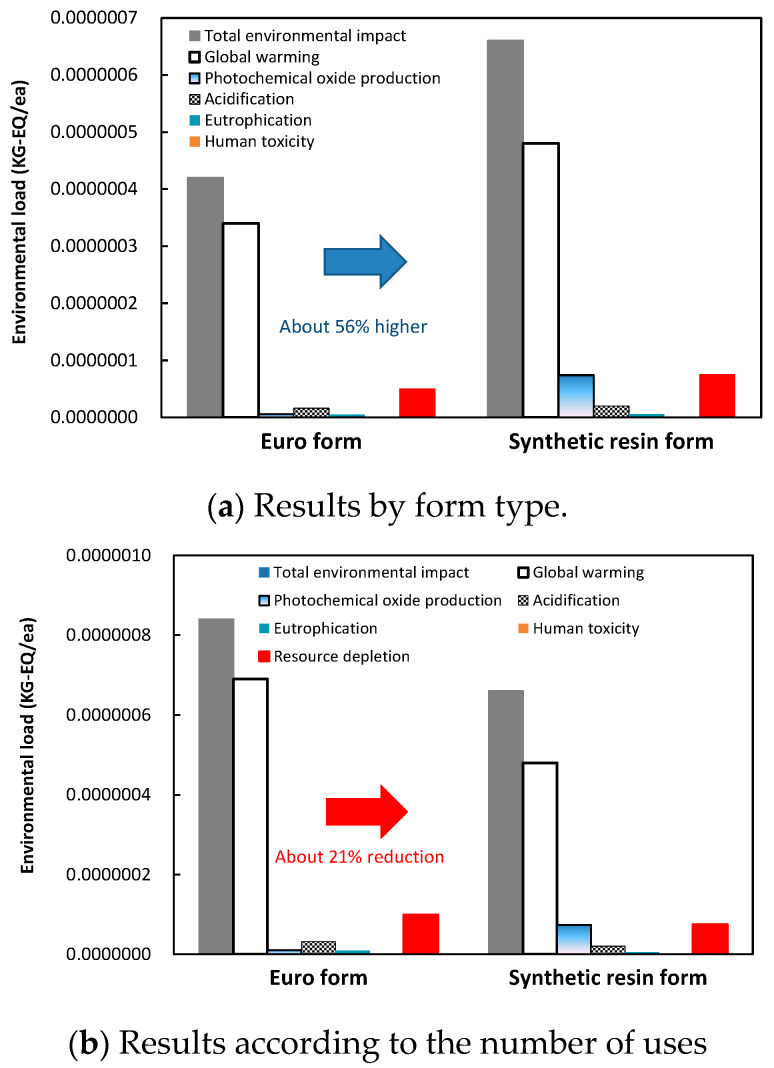
Environmental impact assessment.

**Table 1 materials-16-00696-t001:** Physical and chemical properties of HDPE.

Item	Test Specification	Unit	Test Value
Density	ASTM D792	g/cm^3^	0.961
MFR (190 °C, 2.16 kg)	ASTM D1238	g/10 min	5.5
Softening point (vicat)	ASTM D1525	°C	125
Tensile strength at yield point	ASTM D638	kg/cm^2^	290
Elongation at break	ASTM D638	%	>500
Shore hardness (Shore D)	ASTM D2240	-	65
Flexural modulus 1% secant	ASTM D256	kg·cm/cm	8

**Table 2 materials-16-00696-t002:** Experimental plan.

Specimen Composition	Environmental Condition
Material: Forms Size: 1200 mm × 600 mm (horizontal × vertical) Form Thickness: about 60 mm Form Hammer (metal)	Temperature: 9 ± 1 °C Humidity: 51 ± 3% R.H. Wind speed: 0 ± 1 m/s

**Table 3 materials-16-00696-t003:** Noise test results.

Euro form	Maximum noise level [dB (A)]	No. 1	105.4	average 107.3
		No. 2	107.2	
		No. 3	106.6	
		No. 4	108.4	
		No. 5	108.7	
	Background noise level [dB (A)]		55.2	
Synthetic resin form	Maximum noise level [dB (A)]	No. 1	99.2	average 99.7
		No. 2	100.4	
		No. 3	98.2	
		No. 4	102.1	
		No. 5	98.4	
	Background noise level [dB (A)]		55.2	

**Table 4 materials-16-00696-t004:** Summary of LCI DB used for assessing environmental loads of synthetic resin formwork and the Euro form [37].

	Functional Unit	CO_2_	CO	SOX	NOX	NH_3_	Anthracite	Soft Coal	Natural Gas	Crude Oil
Plywood	m^3^	8.01 × 10^2^	Information closed	2.69 × 10^−1^	1.22 × 10^0^	Information closed	Information closed	Information closed	Information closed	Information closed
Press processing (3500 tons)	one time	1.88 × 10^−1^	1.87 × 10^−5^	6.28 × 10^−4^	4.60 × 10^−4^	4.32 × 10^−7^	3.27 × 10^−7^	1.73 × 10^−7^	8.67 × 10^−3^	8.46 × 10^−3^
Section steel	kg	4.19 × 10^−1^	1.05 × 10^−4^	1.19 × 10^−3^	1.35 × 10^−3^	9.57 × 10^−9^	1.29 × 10^−1^	3.44 × 10^−4^	1.49 × 10^−2^	3.85 × 10^−2^
HDPE	kg	1.88 × 10^0^	9.15 × 10^−2^	1.61 × 10^−3^	2.91 × 10^−3^	1.14 × 10^−6^	1.06 × 10^−4^	1.13 × 10^−4^	9.90 × 10^−2^	1.49 × 10^0^
PE injection molding	kg	2.90 × 10^−1^	2.89 × 10^−5^	9.70 × 10^−4^	7.11 × 10^−4^	6.67 × 10^−7^	1.17 × 10^−7^	6.20 × 10^−8^	1.34 × 10^−2^	1.3 × 10^−2^
Truck over 12 tons	kg/kg-km	1.14 × 10^−6^	1.41 × 10^−8^	-	1.19 × 10^−8^	1.86 × 10^−14^	3.12 × 10^−9^	-	1.63 × 10^−8^	3.72 × 10^−7^

**Table 5 materials-16-00696-t005:** Classification and characterization of environmental impact categories.

Impact Category	Inventory	Characterization Coefficient	Normalization Reference
Resource depletion	Anthracite coal	4.61 × 10^−3^/yr	2.49 × 10^4^ g/pr-yr
	Bituminous coal	4.61 × 10^−3^/yr	
	Natural gas	1.671 × 10^−2^/yr	
	Crude oil	2.48 × 10^−2^/yr	
Global warming	CO_2_	1.00 × 10^0^ g CO_2_-eq/g	5.53 × 10^−6^ g CO_2_-eq/pr-yr
Photochemical oxide production	CO	2.70 × 10^2^ g C_2_H_4_-eq/g	1.03 × 10^4^ g C_2_H_4_-eq/pr-yr
	SO_x_	2.80 × 10^−2^ g C_2_H_4_-eq/g	
	NO_x_	4.80 × 10^−2^ g C_2_H_4_-eq/g	
Acidification	SO_x_	1.00 × 10^0^ g SO_2_-eq/g	3.98 × 10^4^ g SO^2-^eq/pr-yr
	NO_x_	7.00 × 10^−1^ g SO_2_-eq/g	
	NH_3_	1.88 × 10^0^ g SO_2_-eq/g	
Eutrophication	NO_x_	1.30 × 10^−1^ g PO_4_^−3^-eq/g	1.31 × 10^4^ g PO_4_^−3^-eq/pr-yr
	NH_3_	3.50 × 10^−1^ g PO_4_^−3^-eq/g	
Human toxicity	SO_x_	9.60 × 10^−2^ g 1.4 DCB-eq/g	1.48 × 10^6^ g 1.4 DCB-eq/pr-yr
	NO_x_	1.20 × 10^0^ g 1.4 DCB-eq/g	

**Table 6 materials-16-00696-t006:** Weighting of environmental impact categories.

Impact Category	Reduction Factor (N_i_/T_i_)	Relative Significance Factor (fi)	Weight (W_i_)
Resource depletion	1.06	0.218	0.231
Global warming	1.05	0.274	0.288
Photochemical oxide production	1.09	0.060	0.065
Acidification	1.05	0.034	0.036
Eutrophication	1.46	0.026	0.038
Human toxicity	1.42	0.074	0.105

**Table 7 materials-16-00696-t007:** Comparison between power consumption of the Euro form and synthetic resin formwork.

Division	Formworks	Inventory for Environmental Load (kg-eq/30 Uses)						
		Total Environmental Impact	Global Warming	Photochemical Oxide Production	Acidification	Eutrophication	Human Toxicity	Resource Depletion
CO_2_ emissions during formwork	Euro form	4.2 × 10^−7^	3.4 × 10^−7^	5.4 × 10^−9^	1.6 × 10^−8^	4.7 × 10^−9^	1.1 × 10^−9^	5.0 × 10^−8^
	synthetic resin form	6.6 × 10^−7^	4.8 × 10^−7^	7.4 × 10^−8^	2.0 × 10^−8^	5.9 × 10^−9^	1.4 × 10^−9^	7.5 × 10^−8^
CO_2_ emissions during formwork (30 uses)	Euro form	8.4 × 10^−7^	6.9 × 10^−7^	1.1 × 10^−8^	3.2 × 10^−8^	9.5 × 10^−9^	2.3 × 10^−9^	9.9 × 10^−8^
	synthetic resin form	6.6 × 10^−7^	4.8 × 10^−7^	7.4 × 10^−8^	2.0 × 10^−8^	5.9 × 10^−9^	1.4 × 10^−9^	7.5 × 10^−8^

## Data Availability

The result of this research can be applied as a new Synthetic Resin Formwork.

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
