# Peer review of "Life Cycle Environmental Impact Assessment and Applicability of Synthetic Resin Formwork"

_materials, 2023, doi:10.3390/ma16020696_

Round 1
Reviewer 1 Report
In general
This manuscript contains fragments that are not related to each other. Hence, it is difficult to understand the scientific novelty of this study. If this manuscript is restructured, it will help to evaluate it.
In Specific
Lines 13-15 (Abstract). It was written “These results show that the use of synthetic resin formwork can reduce material production by half compared to the Euro form and lead to a reduction in CO2 emissions”.
However, this result was stated as an assumption, not as a fact.
Lines 21-131. The introduction contains trivial information that does not allow understanding the scientific novelty of this study. The introduction does not contain information on formwork life cycle assessment in other publications. Lines 21-95. There is no title for this subsection. The subsections “Cautions for selecting formwork materials” and "Method and field of study" should be moved to the section "Methods". The last paragraph of the introduction should contain the purpose of the study. The purpose of the study should follow from the critical analysis that was presented in the introduction. The introduction needs to be rewritten.
The section "Materials and Methods" should contain information about materials and measurement methods. This section should be written in such a way that the reader of the "Materials" (journal) can reproduce these studies.
Lines 187-191. It was written “To measure the noise, the Solo 01dB-Metravib (France) was used as a sound level meter and the CAL02 01dB-Stell (Japan) was used as a sound level calibrator. When an impact was applied to the center of the metal formwork frame with a hammer, the maximum noise level was measured five times at a location 1.5 m in front of the formwork”.
According to these two proposals, the noise includes two components: “a sound level meter” and “a sound level calibrator” and “the maximum noise level was measured five times at a location 1.5 m in front of the formwork.”
What is the difference between a “sound level meter” and a “sound level calibrator”?
Where did the noise measurement protocol come from?
Why doesn't the introduction contain an analysis of noise level studies at construction sites?
Section "3.1.2 Noise Measurement Results" was written twice.
How was the noise frequency domain analysis done?
Why is there no information on the frequency domain analysis of noise at construction sites in the introduction?
The life cycle assessment procedure was not written clearly. All processes and all materials for all alternatives must be described in detail. This section should be written in such a way that it can be easily repeated.
Lines 313-322. Boundary conditions contain many repetitive phrases, such as "regional boundary" and "technical boundary".
The title of Table 4 contains “assessing environmental loads … the Euro form”.
Where is the data on "Euro form"?
Are the data in tables 5 and 6 obtained from the authors of the manuscript or from the literature?
Line 12. For the first time the term "the number of uses" was used in the abstract.
Line 401. For the second time this term "the number of uses" was only used in the result section.
This term is omitted from both the introduction and the methods sections.
Lines 401-407. These phrases “…the CO2 emissions from the synthetic resin formwork were 32% lower than those of the Euro form when the number of uses was considered. The number of uses for the Euro form is less than 15, which is half the number of uses for synthetic resin formwork” contain unclear result.
Lines 416-417. It was written “…The number of uses for the synthetic resin formwork was assumed to be 30 and that for the Euro form was assumed to be 15”.
But ...it was [only] assumed!
Line 457. However, it was written in conclusion “based on 30 uses”
It is not clear how this assumption was converted into one, and it seems to me the most important of the four conclusions.
Lines 421-423. It was written “These results are similar to that of a previous study on CO2 emissions generated from materials [39]”.
However, the reference [39] is not relevant to the current study.
Line 233. "noise fatigue" was introduced in the results section for the first time.
Line 441. In the conclusion section, the term "noise fatigue" is used in the second time.
There is no definition of this term in the introduction, and no methodology for measuring this phenomenon in the methods section.
Author Response
Thank you for your kind review of this paper.
I revised the paper based on the points pointed out. In addition, I have attached a file with the answer about the contents.
Verification please.
thank you.

Reviewer 2 Report
The authors have experimented the applicability of synthetic resin formwork over Euro form. Although, some parameters have been investigated, however, the cost and reusability of synthetic resin formwork not described properly. These issues must be discussed. Also, some durability tests should have been performed and compared with the available formwork types.
Author Response

(The authors gave the same response as above.)

Round 2
Reviewer 1 Report
accept
Reviewer 2 Report
The quantitative comparison of Euro form and synthetic resin formwork is performed here. Although the authors have not performed any durability tests which I proposed to do in my first revision, still the manuscript is improved from its older version. I have no more comments.